# Essential Oils of *Tagetes minuta* and *Lavandula coronopifolia* from Djibouti: Chemical Composition, Antibacterial Activity and Cytotoxic Activity against Various Human Cancer Cell Lines

Fatouma Mohamed Abdoul-Latif [1,*], Abdirahman Elmi [1,2], Ali Merito [1], Moustapha Nour [1], Arnaud Risler [2], Ayoub Ainane [3], Jérôme Bignon [4] and Tarik Ainane [3]

[1] Medicinal Research Institute, Centre d'Etudes et de Recherche de Djibouti, IRM-CERD, Route de l'Aéroport, Haramous B.P. 486, Djibouti
[2] Laboratoire Lorrain de Chimie Moléculaire (L2CM), Université de Lorraine, CNRS, F-54000 Nancy, France
[3] Superior School of Technology of Khenifra, University of Sultan Moulay Slimane, BP 170, Khenifra 54000, Morocco
[4] Institut de Chimie des Substances Naturelles, CNRS UPR 2301, Université Paris-Saclay, Avenue de la Terrasse, 91198 Gif-sur-Yvette, France
* Correspondence: fatoumaabdoulatif@gmail.com

**Abstract:** The chemical composition of the essential oils of two plants (*Tagetes minuta* L. and *Lavandula coronopifolia* L.) harvested from the Day region (in the north of Djibouti) is the subject of this study. The extraction of essential oils was carried out by hydrodistillation, and the average yield was obtained at a rate of approximately 0.25% for *Tagetes minuta* L. and 0.42% for *Lavandula coronopifolia* L. The analyses of these essential oils by gas chromatography coupled with mass spectrometry identified 13 compounds in the essential oil of *Tagetes minuta* L., including dihydrotagetone (20.8%), artemisia (17.9%), (Z)-tagetenone (12.4%), (-)-spathulenol (11.0%) and estragole (9.5%), were obtained as majority compounds, with a percentage of 71.6%. The essential oil of *Lavandula coronopifolia* L. is characterized by the presence of 42 compounds, including cis-caryophyllene (18.9%), dehydronerolidol (12.8%), iso-longifolanone (11.2%), caryophyllene oxide (8.2%), 10-epi-β-eudesmol (7.7%) and humulene (5.1%), were obtained as the majority chemical constituents, with a percentage of 63.9%. The antimicrobial activities of the essential oils at concentrations of 5% were measured against 12 bacterial strains (Gram positive: *Staphylococcus aureus* (ATCC 29213), *Enterococcus faecalis* (ATCC 29212), *Streptococcus agalactiae* (ATCC 27956), *Staphylococcus epidermidis* and *Corynebacterium* sp.; Gram Negative: *Pseudomonas aeruginosa* (ATCC 27853), *Escherichia coli* (ATCC 25922), *Klebsiella pneumoniae* (ATCC 700603), *Acinetobacter baumannii* (ATCC 19606), *Shigella sonnei* (ATCC 9290), *Salmonella enterica* sv. *Typhimurium* (ATCC 13311) and *Enterobacter cloacae*), and the results of in vitro experiments showed inhibitory effects against most strains tested except Staphylococcus aureus, Enterococcus faecalis and Streptococcus agalactiae. Additionally, both oils were tested for their ability to selectively kill 13 human cancer cells (K562, A549, HCT116, PC3, U87-MG, MIA-Paca2, HEK293, NCI-N87, RT4, U2OS, A2780, MRC-5 and JIMT-T1), and the results obtained, according to the values of $IC_{50}$, show the significant activity of two essential oils, particularly on the HCT116 and A2780 lines, which present values between 0.25 μg/mL and 0.45 μg/mL, respectively.

**Keywords:** essential oils; *Tagetes minuta* L.; *Lavandula coronopifolia* L.; antimicrobial activity; cytotoxic activity

## 1. Introduction

Medicinal plants represent one of the sources of medicines for about 80% of the African population [1–4]. The country of Djibouti is a model of the countries of East Africa, with an arid and desert climate. The average rainfall is low, around 250 mm. However, there are

more than 800 listed plant species capable of adaptation in these difficult conditions, and they may be of interest for their medicinal uses [5,6]. The invaluable expertise of traditional healers is a starting point for the pharmacological and phytochemical investigation of natural medicines [7]. Recently, phytochemicals with antimicrobial potential have been widely explored to identify ingredients for potential medicinal applications [8–10].

On the other hand, cancer is characterized by unlimited growth, invasion and cell metastasis and is presently considered a major cause of death worldwide, while benign tumors are self-limiting [11]. Synthetic treatments remain the only option for cancer chemotherapy, but unfortunately, the majority of these synthetic drugs not only kill tumor cells but also attack healthy cells, and, as a result, they generate serious side effects [12]. Therefore, we have to look for urgent, alternative and sustainable solutions with advanced treatment options and properties that allow improved functionality. Recently, natural products such as aromatic and medicinal plants have become among the solutions used, and in particular, the application of essential oils has been used to better understand basic research, in particular, the antimicrobial activities and potential anticancer activity of essential oils [13–15].

*Tagetes minuta* L. is a plant of the Asteraceae family, and this species is an annual native of South America. It is found in South and Central America, Europe, India, Australia, New Zealand and Russia. In Djibouti, it occurs spontaneously in fallow land where it is considered a weed, or along roads and paths; it measures between 60 and 120 cm high, even 2 m, and grows in the rubble. *Tagetes minuta* L. is used by some healers to cure some illnesses such as flu or cough and to cure wounds and allergies [16–18]. The other plant, *Lavandula coronopifolia* L., is the genus Lavandula and includes around thirty species from the countries of the Mediterranean region to East Africa and Arabia. *Lavandula coronopifolia* L. is a small shrub of the Lamiaceae family, the stems of which can reach 1 m in height. The leaves are cut into narrow lobes. The ear is thin, about 20 cm long. Each flower has a sharp bract 0.3–0.5 calyx in length and is corolla blue to lilac, tube-curved at the base and dilated upwards [19–21].

Given the above background, the objective of this study was to extract and determine the chemical composition of two essential oils from *Tagetes minuta* and *Lavandula coronopifolia* and to evaluate their antibacterial activities against: *Staphylococcus aureus*, *Enterococcus faecalis*, *Streptococcus agalactiae*, *Staphylococcus epidermidis* and *Corynebacterium* sp., *Pseudomonas aeruginosa*, *Escherichia coli*, *Klebsiella pneumoniae*, *Acinetobacter baumannii*, *Shigella sonnei*, *Salmonella enterica* sv. *Typhimurium* and *Enterobacter cloacae* and thus to examine their potential for cytotoxicity against 13 human cancer cells: K562, A549, HCT116, PC3, U87-MG, MIA-Paca2, HEK293, NCI-N87, RT4, U2OS, A2780, MRC-5 and JIMT-T1.

## 2. Material and Methods

### 2.1. Collection of Plants

Samples of the aerial part (stems, leaves and flowers) of *Tagetes minuta* L. and *Lavandula coronopifolia* L. were collected in the region of Day (in the north of Djibouti) (Table 1). Voucher specimens of two plants were identified by Fatouma Mohamed Abdoul-Latif and deposited at the Medicinal Research Institute of Djibouti (CERD) under the codes TM2019324 and LC2019405 for *Tagetes minuta* and *Lavandula coronopifolia*, respectively.

**Table 1.** Place of collection of the two aromatic and medicinal plants used.

| Name of the Plant | Place of Collection |
|---|---|
| *Tagetes minuta* L. | Day (North Djibouti) |
| *Lavandula coronopifolia* L. | (11°45.18′ N; 42°37.73′ E) |

### 2.2. Extraction of Essential Oils

The essential oils of *Tagetes minuta* L. and *Lavandula coronopifolia* L. were obtained by hydrodistillation of the aerial parts of dried plants (stems, leaves and flowers) in fractions

of 120 g and 70 g, respectively, for a period of 3 h using an extractor of the type Clevenger. The essences less dense than water were collected by simple decantation and dried over anhydrous sodium sulfate ($Na_2SO_4$) before analysis [22].

The yield of essential oil (expressed as a percentage) is calculated by the ratio between the weight of the oil extracted and the weight of the plant material used.

The essential oil yield of *Tagetes minuta* L. and *Lavandula coronopifolia* L. was determined relative to the dry matter, by fractions of 120 g and 70 g, respectively, and dried in an oven for 48 h at 60 °C. The resulting essential oil was stored at a temperature of 4 °C in the dark [23].

$$\text{Yield } (\%) = \left(\frac{V}{ms} \times 100\right) \pm \left(\frac{\Delta V}{ms} \times 100\right) \tag{1}$$

where $V$ = volume of essential oil collected and $\Delta V$ = reading error/ms (mass of plant material in the dry state).

### 2.3. Chemical Compositions

The essential oils of *Tagetes minuta* L. and *Lavandula coronopifolia* L. were analyzed by gas chromatography coupled with mass spectrometry (GC/MS). The analyses of the two essential oils were performed using QP2010-Shimadzu equipment operating in the EI mode at 70 eV. An SLB5 column DB 5 ms (30 m, 0.25 mm film thickness) was employed with a 36 min temperature program of 60–320 at 10 °C/min, followed by a 5 min hold at 320 °C. The injector temperature was 250 °C, the flow rate of the carrier gas (helium) was 1 mL/min, and the split ratio was 1:50. The interval of the scan m/z was between 35 and 900, and the identification of the compounds is based on an individual spectrum comparison of each compound in the Shimadzu NIST08 data [24,25].

### 2.4. Antibacterial Activity

According to the joint recommendations of the "Comité de l'Antibiogramme de la Société Française de Microbiologie" (CA-SFM) and the "Clinical and Laboratory Standards Institute" (CLSI, formerly "National Committee for Clinical Laboratory Standards" or NCCLS) [25,26], 12 strains of Reference bacteria were used for our study of Gram-positive bacteria: *Staphylococcus aureus* ATCC 29213, *Enterococcus faecalis* ATCC 29212, *Streptococcus agalactiae* ATCC 27956, *Staphylococcus epidermidis* and *Corynebacterium* sp. The Gram-negative bacteria studied were as follows: *Pseudomonas aeruginosa* ATCC 27853, *Escherichia coli* ATCC 25922, *Klebsiella pneumoniae* ATCC 700603, *Acinetobacter baumannii* ATCC 19606, *Shigella sonnei* ATCC 9290, *Salmonella enterica* sv. *Typhimurium* ATCC 13311 and *Enterobacter cloacae*.

Conditions of the tests carried out were as follows:

- In order to perform antimicrobial testing, each of the two essential oils was tested at concentrations of 5 μL.
- Screening concentration: 5 μL of sample + 95 μL of Mueller–Hinton cation-adjusted medium (CAMHB) = 5% final.
- The bacterial inoculum was carried out in order to have a final concentration of $5 \times 10^5$ [$2 \times 10^5$–$8 \times 10^5$] CFU/mL in the wells.
- In the plate plan, each condition contained a sterile control (medium only), product (medium + final 5% sample) and growth control (bacteria + medium, without sample).
- The growth was detected after incubation in 96-well microplates for 24 h at 35 °C by visualization of a growth cloud/pellet and comparison with the corresponding product control. A microplate corresponded to the test on a bacterium. Only one test/bacteria/product was performed.

### 2.5. Cytotoxicity Tests

The in vitro cytotoxic activities of the essential oils of *Tagetes minuta* L. and *Lavandula coronopifolia* L. were measured against 13 human cell lines (K562, A549, HCT116, PC3,

U87-MG, MIA-Paca2, HEK293, NCI-N87, RT4, U2OS, A2780, MRC-5 and JIMT-T1) by determining concentrations of 50% inhibition of cell proliferation. [27,28].

Cancer cell lines were obtained from the American Type Culture Collection (ATCC, Rockville, MD, USA) or the German Collection of Microorganisms and Cell Cultures of the Leibniz Institute (DSMZ, Braunschweig, Germany) or the European Collection of Cell Cultures (ECACC, Salisbury, UK). Cancer cell lines were grown according to the supplier's instructions.

Human colorectal carcinoma HCT-116, bladder RT4 and bone osteosarcoma U2OS were cultured in Gibco McCoy's 5A supplemented with 10% fetal calf serum (FCS) and 1% glutamine. Lung carcinoma A549, ovarian carcinoma A2780, myeloid leukemia K562, embryonic kidney cells HEK293, gastric carcinoma NCI-N87 and prostate carcinoma PC3 cells were cultured in Gibco RPMI 1640 medium supplemented with 10% of fetal calf serum (FCS) and 1% glutamine. JIMT-T1 breast carcinoma, U87-MG cerebral glioblastoma, Mia-Paca2 carcinoma and MRC5 human lung cells were cultured in Gibco DMEM medium supplemented with 10% fetal calf serum (FCS) and 1% glutamine. All cell lines were maintained at 37 °C in a humidified atmosphere containing 5% $CO_2$.

Cell viability was determined by a luminescence assay according to the manufacturer's instructions (Promega, Madison, WI, USA). For $IC_{50}$ determination, the cells were seeded in 96-well plates ($3 \times 10^3$ cells/well) containing 90 µL of growth medium. After 24 h of culture, the cells were treated with the tested compounds at 8 different final concentrations (10; 5; 1; 0.5; 0.1; 0.05; 0.01 and 0.005 µg/mL). Each concentration was obtained from serial dilutions in culture medium starting from the stock solution. Control cells were treated with the vehicle. Experiments were performed in triplicate.

After 72 h of incubation, 100 µL of CellTiter Glo Reagent was added for 15 min before recording luminescence with a spectrophotometric plate reader PolarStar Omega (BMG LabTech, Ortenberg, Germany). The dose–response curves were plotted with Graph Prism software and the $IC_{50}$ values were calculated using the Graph Prism software from polynomial curves (four or five-parameter logistic equations).

### 2.6. Statistical Analysis

The statistical analysis of the values obtained from the determination of yields, antibacterial activity and cytotoxicity were carried out by Type A evaluation of standard uncertainty with Student's *t* test (*t* < 0.05).

## 3. Results

### 3.1. Essential Oils: Yields and Compositions

After extraction, the average yield of essential oil for each species was calculated based on the dry plant matter obtained from the aerial parts (stems, leaves and flowers) of the plants studied. The essential oil yields obtained are given in Table 2.

**Table 2.** Yields of essential oils.

| Species | Yield (%) |
|---|---|
| *Tagetes minuta* L. | 0.25 ± 0.05 |
| *Lavandula coronopifolia* L. | 0.42 ± 0.06 |

### 3.2. Chemical Compositions

The results of the gas chromatographic analysis coupled with the mass spectrometry of the essential oils of the plants studied are shown in (Table 3). Chromatographic analyses of essential oils made it possible to identify 13 compounds, which represents 100% for *Tagetes minuta* L., and for *Lavandula coronopifolia* L. 42 compounds, which represents approximately 99.4%.

**Table 3.** Chemical composition of the essential oils of *Tagetes minuta* L. and *Lavandula coronopifolia* L.

| Pic | RT | Compounds | *Tagetes minuta* **L.** | *Lavandula coronopifolia* **L.** |
|---|---|---|---|---|
| 1 | 4.39 | α-Pinene | - | 0.7 |
| 2 | 7.10 | D-Limonene | 5.8 | 0.4 |
| 3 | 8.75 | Linalol | - | 0.3 |
| 4 | 9.54 | Isopinocarveol | - | 0.4 |
| 5 | 9.64 | Verbenol | - | 0.9 |
| 6 | 9.67 | Camphor | 1.8 | - |
| 7 | 10.57 | Myrtenol | - | 0.4 |
| 8 | 10.62 | Estragole | 9.5 | - |
| 9 | 10.78 | D-Verbenone | - | 0.2 |
| 10 | 11.18 | Verbenone | 5.4 | - |
| 11 | 11.46 | (Z)-Tagetenone | 12.4 | - |
| 12 | 11.55 | Piperitone | 5.1 | - |
| 13 | 11.99 | Bornyl Acetate | - | 0.6 |
| 14 | 12.13 | 2-Undecanone | 2 | 0.5 |
| 15 | 12.91 | α-Terpineol Acetate | - | 0.3 |
| 16 | 13.32 | Copaene | - | 0.2 |
| 17 | 13.47 | Cubenene | - | 1.5 |
| 18 | 13.67 | Zingiberene | - | 0.2 |
| 19 | 13.83 | Benzene, 2-tert-butyl-1,4-dimethoxy- | 3.3 | - |
| 20 | 13.93 | Cis-Caryophyllene | - | 18.9 |
| 21 | 14.09 | α-Bergamotene | - | 2.6 |
| 22 | 14.36 | β-Sesquiphellandrene | - | 0.8 |
| 23 | 14.41 | Humulene | - | 5.1 |
| 24 | 14.47 | Aromadendrene | - | 0.1 |
| 25 | 14.74 | Germacrene D | - | 0.6 |
| 26 | 14.86 | β-Eudesmene | - | 0.5 |
| 27 | 14.94 | Methyl (2E)-2-nonenoate | - | 0.8 |
| 28 | 15.06 | Bisabolene | - | 0.8 |
| 29 | 15.15 | τ-Cadinene | - | 0.9 |
| 30 | 15.20 | δ-Cadinene | - | 0.7 |
| 31 | 15.24 | α-Panasinsen | - | 0.9 |
| 32 | 15.46 | α-Caryophyllene | - | 0.8 |
| 33 | 15.59 | β- Elemol | - | 0.7 |
| 34 | 15.65 | Epiglobulol | - | 3.6 |
| 35 | 15.71 | Dehydronerolidol | - | 12.8 |
| 36 | 15.81 | Farnesyl acetone | - | 0.2 |
| 37 | 15.95 | (-)—Spathulenol | 11.0 | 0.4 |
| 38 | 16.03 | Caryophyllene Oxide | - | 8.2 |

**Table 3.** *Cont.*

| Pic | RT | Compounds | *Tagetes minuta* L. | *Lavandula coronopifolia* L. |
|-----|-----|-----------|---------------------|-------------------------------|
| 39 | 16.09 | 12-Heptadecyn-1-ol | - | 0.5 |
| 40 | 16.28 | Longipinane | - | 3.4 |
| 41 | 16.37 | Humulene Epoxide | - | 1.5 |
| 42 | 16.41 | Cubenol | 3.3 | - |
| 43 | 16.63 | α-Cadinol | - | 0.5 |
| 44 | 16.72 | τ-Cadinol | 1.7 | 2.8 |
| 45 | 16.89 | 10-epi-β-eudesmol | - | 7.7 |
| 46 | 16.97 | Viridiflorol | - | 3.7 |
| 47 | 17.2 | α-Bisabolol | - | 1.9 |
| 48 | 17.57 | Isolongifolanone | - | 11.2 |
| 49 | 18.7 | Trans-Farnesol | - | 0.2 |
| 50 | 21.57 | Dihydrotagetone | 20.8 | - |
| 51 | 21.96 | Artemisia | 17.9 | - |
| | | Total (%) | 100 | 99.4 |

RT: Retention time.

Analysis of the results given in Table 3 showed that 3-Methyl-2-butenoic acid, cyclobutyl ester (20.8%), artemisia (17.9%), p-Ment- 6-en-2-one (12.4%), (-)-spathulenol (11%) and estragole (9.5%) were obtained as major compounds, with a percentage of 71.6% in Le essential oil of *Tagetes minuta* L. The essential oil of *Lavandula coronopifolia* L. is characterized by the presence of cis-caryophyllene (18.9%), Dehydronerolidol (12.8%), Isolongifolanone (11.2%), Caryophyllene oxide (8.2%), 10-epi-β-eudesmol (7.7%) and Humulene (5.1%) as main chemical constituents, with a percentage of 63.9%.

### 3.3. Antibacterial Activity

Table 4 represent the results of the antibacterial activities of the essential oils of *Tagetes minuta* L. and *Lavandula coronopifolia* L. at 5%.

**Table 4.** Antibacterial activity of the essential oils of *Tagetes minuta* L. and *Lavandula coronopifolia* L.

| Bacterial Strain | *Tagetes minuta* L. | *Lavandula coronopifolia* L. |
|------------------|---------------------|-------------------------------|
| *Staphylococcus aureus* | - | - |
| *Enterococcus faecalis* | - | - |
| *Streptococcus agalactiae* | - | - |
| *Staphylococcus epidermidis* | - | + |
| *Corynebacterium* sp. | + | + |
| *Pseudomonas aeruginosa* | + | + |
| *Escherichia coli* | - | + |
| *Klebsiella pneumoniae* | + | + |
| *Acinetobacter baumannii* | - | + |
| *Shigella sonnei* | + | + |
| *Salmonella enterica* sv. Typhimurium | + | + |
| *Enterobacter cloacae* | - | + |

+: Growth observed, no inhibition; -: no visible growth, inhibition.

According to Table 4, the essential oil of *Tagetes minuta* L. promotes the absence of visible growth against *Staphylococcus aureus* ATCC 29213, *Enterococcus faecalis* ATCC 29212, *Streptococcus agalactiae* ATCC 27956, *Staphylococcus epidermidis, Escherichia coli* ATCC 25922, *Acinetobacter baumannii* ATCC 19606 and *Enterobacter cloacae,* and therefore, possesses inhibitory antimicrobial activities. On the other hand, growth was observed in *Corynebacterium* sp., *Pseudomonas aeruginosa* ATCC 27853, *Klebsiella pneumoniae* ATCC 700603, *Shigella sonnei* ATCC 9290 and *Salmonella enterica* sv. *Typhimurium* ATCC 13311.

The essential oil of *Lavandula coronopifolia* L. shows only inhibitory antimicrobial activity against *Staphylococcus aureus* ATCC 29213, *Enterococcus faecalis* ATCC 29212 and *Streptococcus agalactiae* ATCC 27956, while *Staphylococcus epidermidis, Corynebacterium* sp., *Pseudomonas aeruginosa* ATCC 27853, *Escherichia coli* ATCC 25922, *Klebsiella pneumoniae* ATCC 700603, *Acinetobacter baumannii* ATCC 19606, *Shigella sonnei* ATCC 9290, *Salmonella enterica* sv. *Typhimurium* ATCC 13311 and *Enterobacter cloacae* observed growth.

### 3.4. Cytotoxic Activity

The essential oils of *Tagetes minuta* L. and *Lavandula coronopifolia* L. were evaluated against 13 human cell lines at different concentrations: 10; 5; 1; 0.5; 0.1; 0.05; 0.01 and 0.005 µg/mL. The results, expressed as a percentage of the control, are shown in Tables 5 and 6, and the relative cytotoxicity curves are shown in Figures 1 and 2.

**Table 5.** Results of cytotoxicity activity of essential oils of *Tagetes minuta* L.

| Concentration (µg/mL) | 10 | 5 | 1 | 0.5 | 0.1 | 0.05 | 0.01 | 0.005 |
|---|---|---|---|---|---|---|---|---|
| | 0.132 | 0.135 | 60.143 | 93.914 | 99.212 | 100.686 | 95.323 | 94.160 |
| % Viability of K562 | 0.062 | 0.060 | 58.816 | 102.310 | 101.356 | 103.539 | 97.953 | 99.371 |
| | 0.035 | 0.000 | 55.202 | 95.936 | 95.592 | 101.359 | 94.368 | 98.047 |
| | 0.070 | 21.672 | 46.632 | 72.479 | 85.664 | 98.925 | 101.021 | 95.272 |
| % Viability of A549 | 0.128 | 19.870 | 57.755 | 68.532 | 71.567 | 102.238 | 96.558 | 96.213 |
| | 0.103 | 22.101 | 68.277 | 77.959 | 67.023 | 82.742 | 85.453 | 98.795 |
| | 0.061 | 0.035 | 29.727 | 41.375 | 91.394 | 89.984 | 91.810 | 97.379 |
| % Viability of HCT116 | 0.042 | 0.010 | 35.671 | 35.351 | 88.193 | 86.781 | 93.360 | 100.727 |
| | 0.021 | 0.029 | 41.157 | 42.025 | 90.718 | 85.432 | 99.987 | 97.632 |
| | −0.002 | 0.123 | 68.782 | 74.298 | 92.549 | 106.315 | 110.648 | 112.465 |
| % Viability of MRC5 | 0.004 | 0.156 | 68.729 | 78.604 | 89.021 | 93.533 | 98.798 | 101.784 |
| | 0.011 | −0.002 | 70.030 | 78.168 | 88.810 | 94.511 | 98.705 | 103.778 |
| | 0.022 | 0.030 | 86.001 | 93.193 | 93.199 | 92.794 | 103.045 | 97.490 |
| % Viability of PC3 | 0.078 | 6.858 | 85.724 | 97.571 | 97.431 | 103.990 | 97.318 | 100.992 |
| | 0.050 | 0.161 | 84.679 | 95.804 | 104.181 | 101.358 | 103.014 | 96.191 |
| | 0.036 | 0.108 | 48.441 | 66.427 | 88.847 | 94.622 | 99.669 | 94.643 |
| % Viability of U87 | 0.065 | 0.079 | 54.432 | 69.890 | 87.530 | 96.875 | 106.790 | 102.203 |
| | 0.043 | 0.122 | 60.487 | 78.919 | 98.733 | 102.138 | 93.239 | 104.190 |
| | 0.046 | 3.995 | 77.181 | 95.899 | 98.964 | 99.721 | 96.328 | 95.370 |
| % Viability of MiaPaca | 0.019 | 0.893 | 75.354 | 99.208 | 95.767 | 92.299 | 90.759 | 100.002 |
| | 0.035 | 1.978 | 80.685 | 88.753 | 98.890 | 101.399 | 100.071 | 93.353 |
| | 0.001 | 0.025 | 51.769 | 73.777 | 104.911 | 102.225 | 100.426 | 96.442 |
| % Viability of HEK293 | 0.054 | 0.128 | 79.862 | 80.507 | 99.485 | 97.591 | 106.593 | 101.172 |
| | 0.012 | 0.050 | 61.012 | 73.738 | 105.982 | 104.230 | 103.431 | 98.831 |

**Table 5.** *Cont.*

| Concentration (µg/mL) | 10 | 5 | 1 | 0.5 | 0.1 | 0.05 | 0.01 | 0.005 |
|---|---|---|---|---|---|---|---|---|
| | 0.144 | 0.053 | 67.901 | 78.976 | 84.186 | 83.492 | 102.387 | 99.272 |
| % Viability of NCI-N87 | 0.041 | 0.114 | 76.855 | 82.461 | 88.753 | 94.142 | 101.383 | 99.423 |
| | 0.027 | 0.119 | 69.745 | 78.486 | 90.804 | 95.817 | 100.596 | 102.297 |
| | 0.031 | 0.078 | 69.821 | 104.428 | 97.508 | 105.900 | 101.479 | 99.404 |
| % Viability of RT4 | 0.091 | 0.154 | 79.391 | 87.859 | 93.833 | 94.920 | 94.130 | 101.905 |
| | 0.034 | 0.014 | 71.182 | 84.068 | 91.313 | 97.905 | 92.921 | 96.357 |
| | 0.010 | 0.055 | 56.720 | 88.899 | 99.278 | 107.134 | 91.143 | 93.098 |
| % Viability of U2OS | 0.032 | 0.114 | 61.351 | 91.029 | 98.685 | 102.890 | 100.645 | 96.322 |
| | 0.022 | 0.057 | 61.282 | 86.819 | 93.922 | 106.523 | 96.788 | 94.170 |
| | 0.001 | 0.003 | 17.720 | 37.224 | 88.362 | 91.870 | 91.574 | 97.886 |
| % Viability of A2780 | 0.013 | 0.028 | 16.241 | 29.937 | 91.668 | 96.027 | 99.377 | 108.071 |
| | 0.021 | 0.018 | 16.723 | 38.288 | 98.740 | 99.503 | 104.114 | 103.544 |
| | 0.065 | 11.958 | 69.134 | 87.242 | 95.128 | 91.780 | 84.795 | 89.711 |
| % Viability of JIMT-T1 | 0.103 | 28.375 | 71.067 | 91.623 | 93.433 | 89.443 | 89.379 | 94.941 |
| | 0.074 | 18.390 | 72.015 | 100.651 | 102.194 | 102.049 | 90.739 | 93.169 |

**Table 6.** Results of cytotoxicity activity of essential oils of *Lavandula coronopifolia* L.

| Concentration (µg/mL) | 10 | 5 | 1 | 0.5 | 0.1 | 0.05 | 0.01 | 0.005 |
|---|---|---|---|---|---|---|---|---|
| | 0.034 | 0.109 | 36.511 | 56.768 | 75.124 | 95.390 | 101.671 | 99.787 |
| % Viability of K562 | 0.051 | 0.469 | 38.714 | 61.290 | 81.839 | 107.163 | 106.360 | 100.520 |
| | 0.032 | 1.833 | 46.734 | 62.133 | 78.379 | 97.406 | 94.364 | 97.321 |
| | 0.024 | 1.271 | 24.652 | 68.050 | 68.765 | 85.077 | 89.963 | 94.588 |
| % Viability of A549 | 0.059 | 6.054 | 41.102 | 69.920 | 77.828 | 89.762 | 89.786 | 93.612 |
| | 0.085 | 12.962 | 48.592 | 72.926 | 79.527 | 99.723 | 97.474 | 100.101 |
| | 0.023 | 0.141 | 13.049 | 51.141 | 58.972 | 77.710 | 96.775 | 98.245 |
| % Viability of HCT116 | 0.050 | 0.584 | 10.309 | 44.107 | 72.297 | 88.930 | 101.873 | 101.182 |
| | 0.020 | 0.558 | 9.474 | 46.128 | 72.595 | 88.489 | 99.253 | 104.028 |
| | 0.015 | 1.865 | 3.253 | 28.387 | 51.198 | 71.540 | 79.800 | 96.299 |
| % Viability of MRC5 | 0.009 | 0.838 | 1.147 | 22.385 | 52.534 | 68.868 | 92.061 | 93.101 |
| | 0.042 | 2.075 | 2.510 | 26.689 | 52.994 | 70.290 | 95.944 | 104.164 |
| | 0.067 | 6.956 | 51.243 | 77.226 | 93.769 | 96.101 | 103.849 | 97.692 |
| % Viability of PC3 | 0.050 | 7.659 | 45.384 | 72.479 | 83.384 | 93.532 | 99.417 | 103.838 |
| | 0.054 | 8.462 | 42.819 | 75.278 | 85.505 | 103.585 | 98.788 | 99.743 |
| | 0.041 | 0.125 | 18.368 | 31.601 | 77.022 | 93.789 | 94.011 | 98.784 |
| % Viability of U87 | 0.102 | 0.186 | 23.654 | 41.690 | 83.753 | 91.028 | 97.690 | 106.968 |
| | 0.048 | 0.301 | 24.526 | 40.512 | 89.628 | 88.848 | 104.643 | 98.310 |
| | 0.037 | 0.055 | 2.898 | 42.625 | 86.300 | 90.164 | 92.980 | 91.068 |
| % Viability of MiaPaca | 0.037 | 0.086 | 8.401 | 42.255 | 82.446 | 82.187 | 93.025 | 94.326 |
| | 0.046 | 0.126 | 17.029 | 45.138 | 83.484 | 93.773 | 100.560 | 101.937 |

**Table 6.** *Cont.*

| Concentration (µg/mL) | 10 | 5 | 1 | 0.5 | 0.1 | 0.05 | 0.01 | 0.005 |
|---|---|---|---|---|---|---|---|---|
| % Viability of HEK293 | 0.019 | 0.063 | 15.112 | 36.926 | 46.487 | 57.827 | 66.485 | 95.930 |
| | 0.022 | 0.112 | 18.459 | 39.662 | 45.614 | 70.053 | 78.714 | 103.155 |
| | 0.035 | 3.886 | 23.315 | 40.251 | 59.402 | 72.740 | 86.680 | 106.725 |
| % Viability of NCI-N87 | 0.367 | 44.866 | 85.526 | 87.160 | 91.126 | 97.501 | 98.440 | 95.893 |
| | 0.211 | 44.373 | 74.834 | 80.070 | 94.241 | 97.447 | 102.724 | 98.559 |
| | 0.167 | 50.366 | 81.310 | 87.011 | 97.033 | 96.259 | 98.942 | 100.290 |
| % Viability of RT4 | 0.071 | 22.161 | 47.684 | 74.115 | 87.597 | 101.157 | 103.300 | 97.422 |
| | 0.131 | 20.319 | 59.058 | 70.079 | 73.182 | 104.545 | 98.736 | 98.384 |
| | 0.105 | 22.600 | 69.818 | 79.718 | 68.535 | 84.609 | 87.381 | 101.024 |
| % Viability of U2OS | 0.051 | 6.454 | 59.222 | 83.786 | 107.296 | 100.329 | 104.358 | 109.611 |
| | 0.100 | 16.656 | 57.771 | 83.592 | 94.799 | 89.068 | 98.967 | 99.516 |
| | 0.107 | 18.563 | 51.549 | 80.135 | 96.574 | 82.410 | 97.473 | 101.379 |
| % Viability of A2780 | 0.033 | 0.328 | 18.764 | 33.397 | 64.800 | 78.758 | 86.224 | 100.504 |
| | 0.038 | 0.275 | 17.789 | 32.112 | 64.196 | 74.316 | 86.299 | 98.366 |
| | 0.050 | 0.895 | 21.274 | 36.969 | 58.933 | 77.973 | 90.253 | 97.535 |
| % Viability of JIMT-T1 | 0.075 | 17.771 | 24.119 | 87.348 | 87.320 | 102.472 | 95.573 | 101.509 |
| | 0.075 | 19.637 | 22.148 | 86.709 | 85.966 | 102.907 | 97.682 | 108.105 |
| | 0.092 | 21.730 | 19.268 | 92.033 | 96.284 | 110.004 | 100.831 | 118.819 |

From the analysis of the dose–response curves, $IC_{50}$s were determined, and we observed that the essential oils of *Tagetes minuta* L. and *Lavandula coronopifolia* L. exhibited cytotoxicity towards all the cancer cell lines (Table 7).

**Table 7.** $IC_{50}$ values for essential oils of *Tagetes minuta* L. and *Lavandula coronopifolia* L.

| | Inhibition Concentration at 50% ($IC_{50}$ in µg/mL) | | | | | |
|---|---|---|---|---|---|---|
| Cell Line | *Tagetes minuta* L. | *Lavandula coronopifolia* L. | Vinblastine * | Doxorubicine * | Combrestatin A4 * | Monomethyl Auristatin E * |
| K562 | 1.06 ± 0.05 | 0.67 ± 0.15 | 20.00 ± 0.12 | - | 5.0 ± 0.3 | 3.12 ± 0.2 |
| A549 | 1.57 ± 0.73 | 0.92 ± 0.14 | - | 56.6 ± 0.84 | 20.0 ± 0.1 | 0.46 ± 0.05 |
| HCT116 | 0.47 ± 0.04 | 0.25 ± 0.03 | 35.00 ± 0.84 | - | 2.0 ± 0.1 | 2.07 ± 0.02 |
| PC3 | 1.71 ± 0.18 | 0.97 ± 0.07 | - | 2.09 ± 0.03 | | 0.36 ± 0.03 |
| U87-MG | 1.01 ± 0.12 | 0.34 ± 0.04 | 2.00 ± 0.04 | 99.61 ± 2.34 | 9.0 ± 0.5 | 0.21 ± 0.03 |
| MIA-Paca2 | 1.61 ± 0.06 | 0.45 ± 0.06 | - | - | - | 4.36 ± 0.2 |
| HEK293 | 1.20 ± 0.32 | 0.12 ± 0.05 | - | - | - | - |
| NCI-N87 | 1.48 ± 0.10 | 4.22 ± 1.38 | - | - | - | 1.65 ± 0.07 |
| RT4 | 1.37 ± 0.29 | 1.57 ± 0.73 | - | 36.29 ± 1.20 | - | 0.5 ± 0.01 |
| U2OS | 1.16 ± 0.04 | 1.28 ± 0.14 | - | - | - | - |
| A2780 | 0.36 ± 0.05 | 0.21 ± 0.01 | - | - | - | 0.45 ± 0.01 |
| MRC-5 | 1.33 ± 0.14 | 0.12 ± 0.01 | - | 39.88 ± 1.22 | - | - |
| JIMT-T1 | 2.13 ± 0.38 | 0.71 ± 0.03 | - | - | - | - |

(*) reference of test.

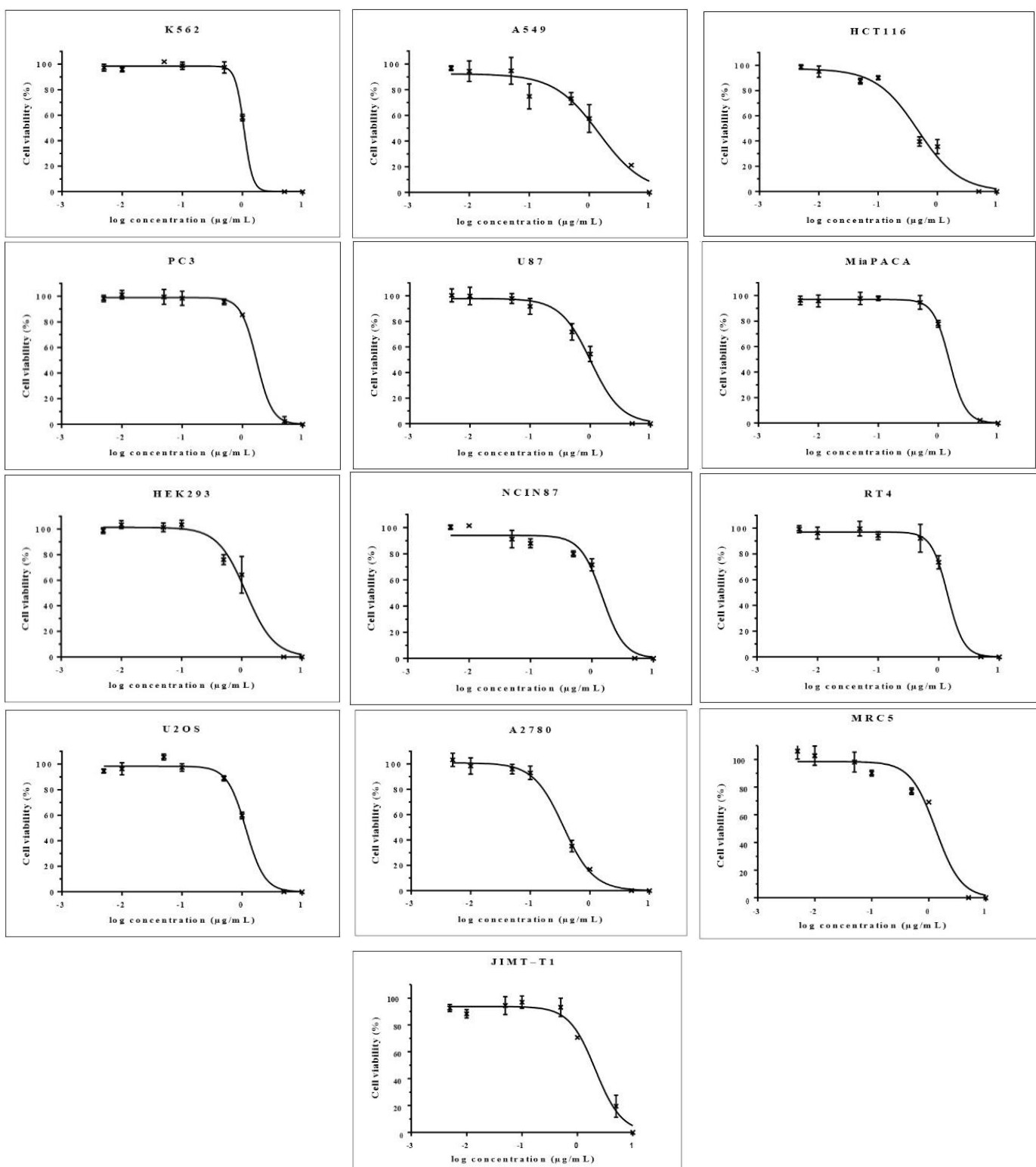

**Figure 1.** Cytotoxicity curves of the essential oil of *Tagetes minuta* L.

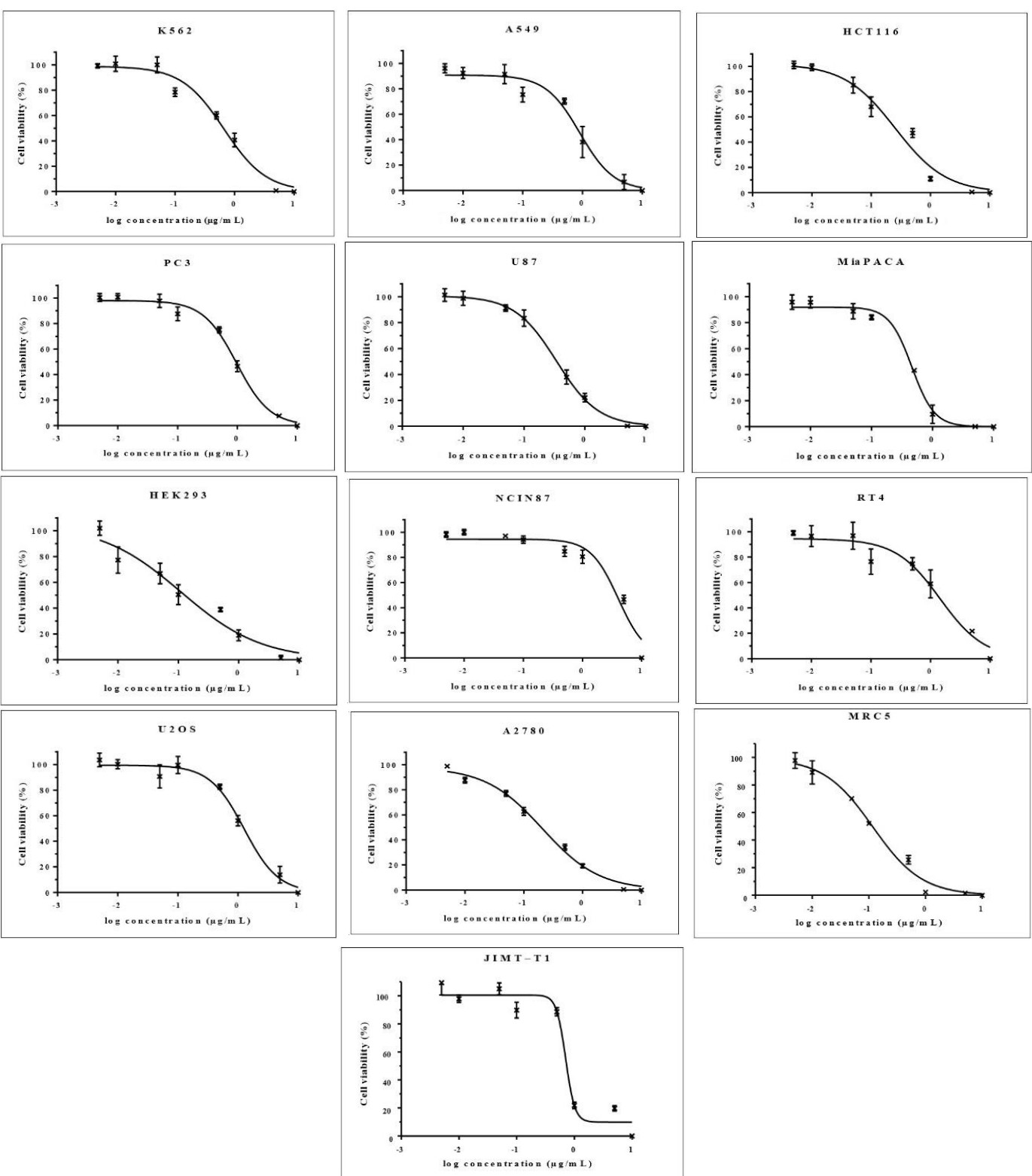

**Figure 2.** Cytotoxicity curves of the essential oil of *Lavandula coronopifolia* L.

## 4. Discussion

Natural products are well recognized for their medical and pharmaceutical benefits and their minimal side effects. *Tagetes minuta* and *Lavandula coronopifolia* are some of these medicinal plants traditionally known throughout the country of Djibouti. The anticancer effect of *Tagetes* and *Lavandula* genera for the treatment of pathogenic diseases and cancer

have been reported [29,30]. In the present study, the chemical composition, antibacterial activities and anticancer potential of two essential oils of *Tagetes minuta* and *Lavandula coronopifolia* from Djibouti against 12 bacterial strains and 13 human cancer cell lines were evaluated. The two essential oils obtained by the hydrodistillation method gave yields of 0.25% and 0.42%, respectively, of *Tagetes minuta* and *Lavandula coronopifolia*. The results of the GC–MS chromatographic analysis of this study are in agreement with previous reports, where the major component of the essential oil of *Tagetes minuta* was dihydrotagetone (20.8%), while artemisia (17.9%), (Z)-tagetenone (12.4%), (-)-spathulenol (11.0%) and estragole (9.5%) were present in significant amounts; the major component of the essential oil of *Lavandula coronopifolia* was cis-caryophyllene (18.9%), while other compounds, such as dehydronerolidol (12.8%) and isolongifolanone (11.2%), were present in remarkable secondary amounts.

The results obtained by Chamorro et al. (2008) [31] show the essential oil composition of *Tagetes minuta* L. collected at different places in the province of Chaco (Argentina), and they presented six main components, which represent more than 90% of the essential oil of *Tagetes minuta*. The main components which were identified are β-phelandrene, limonene, β-ocimene, dihydrotagetone, tagetone and tagetenone. The results reported by Pichette et al. (2005) [32] and Chalchat et al. (1995) [33] show that the essential oil composition of *Tagetes minuta* L. from Rwanda is similar to that of North America and Hungary, with the significant presence of cis β-ocimene. The results presented by Chagonda et al. (1999) [34] show that the main oil components of the whole semi-dry plant of *Tagetes minuta* L. from Zimbabwe were dihydrotagetone (2.2–43.3%), (Z)-p-ocirnene (20.7–41.4%) and (Z)-tagetenone (0.3–31.4%). Another study carried out by Rajeswara Rao et al. (2006) [35] on the essential oils of *Tagetes minuta* L. cultivated in India and extracted by hydrodistillation contained the largest quantities of sesquiterpene hydrocarbons, oxygenated sesquiterpenes, myrcene, dihydrotagétone + (E)-β-ocimene, α-thujone, (E)-tagetone, (Z)-ocimenone, bornyl acetate, β-caryophyllene, δ-cadinene, spathulenol and caryophyllene oxide.

Concerning the essential oil of *Lavandula coronopifolia* L., the results obtained by Ait Said et al. (2015) [36] present 29 main components, representing 92.8% of the oil. Oxygenated monoterpenes are the main group of constituents, representing 52.6%. Carvacrol was the main compound, with a percentage of 48.9%, followed by E-caryophyllene (10.8%). In a previous study in Tunisia presented by Messaoud et al. (2012) [37], they show that trans-b-ocimene (26.9%), carvacrol (18.5%) and b-bisabolene (13.1%) were most representative in the essential oil of *Lavandula coronopifolia* L. In another study by Hassan et al. (2014) [38], the essential oil of *Lavandula coronopifolia* L. was collected in Saudi Arabia and showed 46 components, including the main components phenol-2-amino-4,6-bis (1,1-dimethylethyl) (51.18%) and carvacrol (4.35%). The different variations observed in the relative amounts of the main components of the two essential oils can be explained by variations in geographical and seasonal positions, climatic changes, genotypes, growing conditions and even extraction conditions and methods.

Regarding the microbial evaluations of the two essential oils of *Tagetes minuta* and *Lavandula coronopifolia*, the tests were carried out at concentrations of 5% against 12 bacterial strains, from which the results show interesting activities against *Corynebacterium* sp., *Pseudomonas aeruginosa*, *Klebsiella pneumoniae*, *Shigella sonnei* and *Salmonella enterica* sv. *Typhimurium*. In addition, the essential oil of *Lavandula coronopifolia* showed interesting activity against *Staphylococcus epidermidis*, *Escherichia coli*, *Acinetobacter baumannii* and *Enterobacter cloacae*; in addition, the latter is resistant to the essential oil of *Tagetes minuta*. The other strains, such as *Staphylococcus aureus*, *Enterococcus faecalis* and *Streptococcus agalactiae*, did not show any activity, which proves their resistance to the products tested.

It is noted that the antibacterial activities of the two essential oils of *Tagetes minuta* and *Lavandula coronopifolia* have been the subject of several scientific works according to several in vitro and in vivo methods [39–42]. The antibacterial activity of *Tagetes minuta* according to Walia et al. (2020), from several locations in Himalaya (India) [43], showed the effectiveness of its essential oil against four strains: *Micrococus luteus, Staphylococcus*

*aureus*, *Klebsiella pneumoniae* and *Pseudomonas aeruginosa*, with better activity against Gram (+) bacteria than against Gram (-) bacteria. This hypothesis was already confirmed by Wanzala et al. (2016) during the review about of the bioactive properties of the essential oil of *Tagetes minuta*. The antibacterial activity of *Lavandula coronopifolia* from Palestine, according to Nassef et al. (2022) [44], showed good results for the essential oil against *Proteus vulgaris*, *Klebsiella pneumonia*, *Staphylococcus aureus*, *Pseudomonas aeruginosa*, *Escherichia coli* and clinically diagnosed methicillin-resistant *Staphylococcus aureus* (MRSA). These results confirm our studies on bacterial strains and the good applications for the valorization of two essential oils in the medical field.

Regarding the cytotoxic activities of *Tagetes minuta* and *Lavandula coronopifolia* against the 13 cancer cells: K562, A549, HCT116, PC3, U87-MG, MIA-Paca2, HEK293, NCI-N87, RT4, U2OS, A2780, MRC-5 and JIMT-T1, the results obtained, compared to the low concentrations used for the two essential oils between 0.005 µg/mL and 10 µg/mL, showed good inhibitions of experimental tests of the cell lines. The 50% inhibition concentration ($IC_{50}$) values are encouraging values compared with reference pharmaceutical products such as vinblastine, doxorubicine, combrestatin A4 and monomethyl auristatin E. According to the literature [45,46], cytotoxic examinations of several cancer cell lines have been conducted using several colorimetric, spectroscopic and luminometric methods. The works of Oyenihi et al. (2021) [47] and Nassef et al. (2022) [44], following the tests of extracts of *Tagetes minuta* and the essential oil of *Lavandula coronopifolia*, respectively, on cancerous cell lines by spectrophotometric methods confirmed our results.

## 5. Conclusions

Today the valuation of aromatic and medicinal plants has been introduced in several applications, in particular essential oils which have been increasingly demanded by the cosmetic, food and pharmaceutical industries, highlighting the importance of carrying out in-depth scientific studies on essential oils, not only for their chemical characterization but also for the possibility of relating chemical content to functional properties. In this regard, the methods of evaluation of biological activities are not only methods that highlight aromatic or preservative activities but also are methods with options and advanced processing properties that allow us improved functionalities and could be beneficial for applications, cosmetics, nutraceuticals and pharmaceuticals. Similar to these ideas, we have valued two Djiboutian aromatic and medicinal plants (*Tagetes minuta* L. and *Lavandula coronopifolia* L.). The extraction was carried out by hydrodistillation, with the average yield of essential oils obtained at rates of approximately 0.25% for *Tagetes minuta* L. and 0.42% for *Lavandula coronopifolia* L. Analyses of these essential oils by chromatography have identified 13 compounds, which represents 100% for *Tagetes minuta* L., and for *Lavandula coronopifolia* L. 42 compounds, which represents approximately 99.4%. The evaluation of cytotoxicity activity against 13 human cell lines showed remarkable cytotoxicity depending on the dose used, and the antibacterial activity against 12 strains was variable. On the other hand, the activities studied in this work are probably due to the major constituents of the essential oils or to the cooperation of their components despite it being very difficult to attribute the biological activities of a total essential oil to one or a few active ingredients because an essential oil always contains a mixture of different chemical compounds.

**Author Contributions:** Conceptualization, F.M.A.-L. and T.A.; methodology, A.E.; software, A.M., M.N., A.R., A.A. and J.B.; validation, F.M.A.-L., T.A. and A.A.; formal analysis, A.R. and J.B.; investigation, F.M.A.-L.; resources, F.M.A.-L.; writing—original draft preparation, A.A.; writing—review and editing, F.M.A.-L.; visualization, T.A.; supervision, A.A.; project administration, F.M.A.-L. All authors have read and agreed to the published version of the manuscript.

**Funding:** This research received no external funding.

**Data Availability Statement:** Data is contained within the article.

**Conflicts of Interest:** The authors declare no conflict of interest.

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
