# Peer review of "Essential Oils of Tagetes minuta and Lavandula coronopifolia from Djibouti: Chemical Composition, Antibacterial Activity and Cytotoxic Activity against Various Human Cancer Cell Lines"

_2037-0164, doi:10.3390/ijpb13030026_

Round 1

Reviewer 1 Report

For essential oil extraction the vegetal material was fresh or dried?

Row 192 "Analysis of the results given in Table 4 showed that" ..... the number of the table about you discuss here is table 3

Table 3 - use caps or small letter for the compounds name (is about the first letter).

Check the italic style for plants name

I suggest to use bacterial strains obtained from the patients in order to evaluate the antimicrobial activity.

Present more information about the possible mechanisms by which the essential oils control cancerous cells.

Author Response

Q1. For essential oil extraction the vegetal material was fresh or dried?

The extraction was made according to dried plants. we added the expression in row 93.

Q2. Row 192 "Analysis of the results given in Table 4 showed that" ..... the number of the table about you discuss here is table 3

The  requested correction has been made.

Q3. Table 3 - use caps or small letter for the compounds name (is about the first letter).

The requested corrections have been made (compound initials are capitalized).

Q4. Check the italic style for plants name

We have corrected the italic style for plant names throughout the manuscript.

Q5. I suggest to use bacterial strains obtained from the patients in order to evaluate the antimicrobial activity.

Indeed, we have selected the bacteria according to a theoretical study which deals with the relationship between cancer and some bacterial infections.

Q6. Present more information about the possible mechanisms by which the essential oils control cancerous cells.

For the relationship between essential oils and the control of cancer cells, we are in the process of producing a new article which generates dozens of mechanisms. this remark for us is a line of research that must be developed

Reviewer 2 Report

Page 5, Section 2.1. Collection of plants. It is good to write the voucher specimen used to identify the plants. Describe as precisely as possibly the area where the plants were collected.

Please check once again the all technical issues in the manuscript. It should be written 1 mL/min, 250 °C. You have added spaces that are not necessary.

Page 8 and 9, Tables 5 and 6. Please check the spelling of viabilité.

I think that figure 1 and 2 are repeating the information given in table 7. I would suggest that the authors think if this is necessary.

Page 12, section 4. Discussion. I think that the authors should try to explain in a mechanistic way the antimicrobial and the anticancer activity of the essential oils. They have not performed such analyses, but I believe they could support their hypothesis with some examples from other publications.

Author Response

Q1. Page 5, Section 2.1. Collection of plants. It is good to write the voucher specimen used to identify the plants. Describe as precisely as possibly the area where the plants were collected.

voucher specimen has been added.

the plant collection area already mentioned in Table 1.

Q2. Please check once again the all technical issues in the manuscript. It should be written 1 mL/min, 250 °C. You have added spaces that are not necessary.

The requested changes have been made to the unit spaces.

Q3. Page 8 and 9, Tables 5 and 6. Please check the spelling of viabilité.

we changed the spelling of "Viabilité" to "Viability".

Q4. I think that figure 1 and 2 are repeating the information given in table 7. I would suggest that the authors think if this is necessary.

The information in table 7 gives the 50% inhibition concentrations which are obtained from figures 1 and 2, the latter presenting curves according to a computer model of the graphical representations of all the experimental concentrations as a function of the cell viabilities.

Q5. Page 12, section 4. Discussion. I think that the authors should try to explain in a mechanistic way the antimicrobial and the anticancer activity of the essential oils. They have not performed such analyses, but I believe they could support their hypothesis with some examples from other publications.

Explanations according to other previous articles have been added according to mechanisms of antibacterial activities and cytotoxicity.

Reviewer 3 Report

I have read the manuscript entitled “Essential oils of Tagetes minuta and Lavandula coronopifolia from Djibouti: Chemical composition, antibacterial activity and cytotoxic activity against various human cancer cell lines”. I have the following comments to the authors:  1. Please use a special template for writing a manuscript that can be found in the Instructions for authors. 2. References 19-21 belong to Tagetes minuta and not to Lavandula coronopifolia. 3. From the Introduction it is not clear why Lavandula coronopifolia was chosen for research. Is it used in medicine Djibouti? 4. In the Introduction specify the degree of chemical study of Tagetes minuta and Lavandula coronopifolia.  5. Specify the geographic coordinates of the location where plants were collected.  6. It is necessary to remove the Traditional use from Materials and Methods. 7. Lines 142-146 are suitable for Introduction and not to Materials and Methods.  8. What standard substances were used in biological experiments? Where is this data? 9. Line 212. There is no Table 8 in the text of the manuscript. 10. Table 5 and 6. Why many values are more than 100%? 11. Line 302. Interesting activity - please replace the term.  12. Make a list of references in accordance with the Instructions for authors.     

Author Response

I have read the manuscript entitled “Essential oils of Tagetes minuta and Lavandula coronopifolia from Djibouti: Chemical composition, antibacterial activity and cytotoxic activity against various human cancer cell lines”. I have the following comments to the authors: 

Q1. Please use a special template for writing a manuscript that can be found in the Instructions for authors.

For the template of the article that exists in the journal site is granted according to a computer program that is not valid in our institute softawre.

Q2. References 19-21 belong to Tagetes minuta and not to Lavandula coronopifolia.

bibliographical references have been corrected. corrected references 16-18 (19-21) for Tagetes minuta and added correct references 19 -21 for Lavandula coronopifolia

Q3. From the Introduction it is not clear why Lavandula coronopifolia was chosen for research. Is it used in medicine Djibouti?

The choice of Lavandula coronopifolia was made according to ethnobotanical studies carried out by our institute with the collaboration of the public and private sectors of Djibouti. According to this study Lavandula coronopifolia presents a traditional treatment with other natural substances.

Q4. Specify the geographic coordinates of the location where plants were collected. 

Geographic coordinates have been added in Table 1.

Q5. It is necessary to remove the Traditional use from Materials and Methods.

traditional use deleted from Table 1 of Materials and Methods.

Q6. Lines 142-146 are suitable for Introduction and not to Materials and Methods. 

The objective already mentioned in the introduction, then the first section 2.5. has been changed to better illuminate the test performed.

Q7. What standard substances were used in biological experiments? Where is this data?

For both biological activities have not made standards. the activities were carried out by a certified laboratory in France (Dr. Jérôme BIGNON) who can evaluate the tests according to international standards).

Q8. Line 212. There is no Table 8 in the text of the manuscript.

the requested correction has been made we have replaced ''table 8'' by ''table 4''.

Q9. Table 5 and 6. Why many values are more than 100%?

The values have been more  100% according to several random errors:

- calibration curve errors.

- reading errors.

- cell culture errors.

- the quality of the chemicals used.

-..

Q10. Line 302. Interesting activity - please replace the term. 

We replaced the term ''remarkable”  with the term “Interesting”

Q11. Make a list of references in accordance with the Instructions for authors.

we have made the requested corrections

Round 2

Reviewer 1 Report

No new comments